# Momentary assessment of parent and child emotion regulation to inform the design of a new emotion-focused parenting app

Tomer S. Berkowitz[1]*, John W. Toumbourou[1,2], Subhadra Evans[1,2], Matthew Fuller-Tyszkiewicz[1,2], Elizabeth M. Westrupp[1,2]

**1** School of Psychology, Deakin University, Geelong, Australia, **2** Centre for Social and Early Emotional Development, Deakin University, Geelong, Australia

* tsberkowitz@deakin.edu.au

## Abstract

Parenting programs show strong evidence for improving parent and child mental health, however, their population reach has been extremely low. Online ecological momentary interventions have been shown to benefit reach in adult mental and public health contexts but have not yet been tested in parenting programs. The current study sought to inform a larger project developing an emotion-focused parenting program designed to provide in-the-moment parenting support. Australian parents of children aged 2–4 years ($N = 89$, $M = 36$ years) were recruited online and completed a baseline questionnaire, followed by a 1-minute survey five times per day over a 7-day period. Of 19 ecological momentary assessment items tested, results showed that six items best measured in-the-moment parent and child negative affect and emotion dysregulation based on five purpose-defined criteria: sensitivity, relevance, alignment, frequency, and validity. These items were used to identify parenting situations and times of day associated with heightened parent and child negative affect and emotion dysregulation. Findings showed an overall response rate of 81%, with participant engagement highest at 7:30am and 7:30 pm. Parent/child dysregulation was heightened during evenings and parenting situations elicited differing levels of dysregulation in children and parents. The study provides novel data that will inform how best to develop a parenting program that provides parents and children with flexible, in-the-moment parenting support.

## Introduction

There is strong evidence showing that emotion-focused parenting programs are effective in improving child emotion regulation and reducing mental health problems [1,2]. However, traditional barriers to parenting programs, including in-person delivery, specific locations, and set times [3,4], mean that only a third of parents who are offered parenting programs enrol in them, and up to 60% of parents drop out [5]. The

**Data availability statement:** All data files are available from the Open Science Framework (DOI: 10.17605/OSF.IO/UG93J).

**Funding:** The author(s) received no specific funding for this work.

**Competing interests:** The authors have declared that no competing interests exist.

last two decades have seen a substantial shift towards online programs [6], but have not been associated with improvements in retention and engagement [7], perhaps because they have not been specifically adapted for the online context [8].

Online ecological momentary interventions offer a way to improve engagement by delivering tailored support in the moment when it is most needed [9]. Although these interventions have demonstrated benefits in reach and outcomes in adult mental and public health contexts [10], they have not yet been utilised for parenting programs. To inform the development of a parenting program as part of larger project, this study sought to identify a brief set of items from pre-existing measures suitable for measuring in-the-moment parent and child emotion dysregulation and negative affect. This study also sought to determine when parents were most likely to engage with smartphone app prompts, and which difficult parenting situations and times of day were associated with higher levels of parent and child dysregulation and engagement.

Emotion regulation refers to the ability of individuals to understand emotions and effectively manage the expression of emotions in a socially acceptable manner [11]. A moderate-to-strong evidence base shows that poor emotion regulation skills (i.e., high levels of emotion dysregulation) in childhood are associated with later adverse outcomes in domains such as academic achievement [12], peer relationships [13,14], and behaviour difficulties [15]. Patterns of high emotion dysregulation over childhood are also associated with poor mental health and suicidality in subsequent years [16–18]. Indeed, strong evidence from a meta-analysis (35 studies, combined $N = 12,516$) supports a moderate to large association between adaptive (mean effect size = 0.29–0.51) and maladaptive (mean effect size = 0.21–0.55) emotion regulation strategies and depressive and anxiety symptoms in children aged 11−18 years [19]. Another meta-analysis, conducted by Compas et al [20], showed a moderate negative association between emotion regulation and internalising and externalising behaviours in children aged 5−10 years (19 studies, mean effect size = 0.30–0.32) and a small to moderate negative association in adolescents aged 11−19 years (14 studies, mean effect size = 0.13–0.20). Existing evidence therefore suggests that equipping children and young people with strong emotion regulation skills in early childhood may be an effective means of reducing population-level mental health problems.

Emotion socialisation theory proposes that parenting beliefs and practices, and the family emotional climate, directly affect children's emotional development [21]. This includes the way in which parents understand and regulate their own emotions, the way in which they display emotions, and the way in which they communicate about emotions with their children [22]. In turn, poor parent emotion regulation has been found to be associated with poor child emotion regulation [23,24], leading to calls for programs that focus on improving both parent and child emotion regulation skills [25]. Parents have a particularly strong influence on child development in early childhood, a time during which children are almost solely reliant on their primary caregivers, and consequently play a significant role in shaping children's developing emotion regulation skills [26].

Parenting programs have demonstrated efficacy in improving child developmental outcomes, including emotion regulation skills [27]. England-Mason and Gonzalez

[28] conducted a review of parenting programs anchored in emotion socialisation theory and measuring change in emotion regulation skills of children aged 0–6 years. Of an initial 1,117 articles retrieved for screening, 12 articles identified three distinct intervention frameworks developed for parents of young children, with two progressing past the pilot stage: Tuning in to Kids and Parent-Child Interaction Therapy-Emotion Development. Both programs showed benefits in improving parents' coaching of their children's emotions [28]. Of the two programs, Tuning in to Kids has the strongest evidence for enhancing emotion regulation in parents and children. The program is a two-hour, face-to-face program delivered over a six-week period by a trained facilitator in a group setting, and has been adapted to suit children at different ages (Tuning in to Toddlers, 1–3 years; Tuning in to Kids, 4–10 years; Tuning in to Teenagers, 11–17 years), as well as a program designed specifically for fathers (Fathers Tuning in to Kids), with all variants found to improve parent emotion socialisation [29–31].

The second program, the Parent-Child Interaction Therapy-Emotion Development intervention, posits that depression in young children is caused by poor emotion regulation [32]. A randomised controlled trial comparing an intervention group to a waitlist control group found that, after 20 sessions with a trained facilitator over an 18-week period, parents showed higher levels of emotion-focused parenting techniques after receiving the intervention [33].

Despite evidence of program efficacy, existing parenting programs have had low uptake and high attrition in the general parenting population [4,34,35]. The aforementioned studies, Tuning in to Kids and Parent-Child Interaction Therapy-Emotion Development, reported a 17–20% and 30% attrition rate, respectively [32,36]. This low rate can be partly attributed to barriers to access and perceived burden. Butler et al [3] conducted a systematic review of qualitative studies that analysed parent experiences of parenting programs (26 studies, combined $N = 822$), and found that barriers to parent engagement and retention were often centred around difficulties in physically attending sessions and difficulties in scheduling time to partake in programs. Thus, although programs like Tuning in to Kids result in improvements to parent emotion regulation skills, the original in-person group format relies on multi-week, face-to-face sessions with trained facilitators. For many parents, this requires them to travel, arrange childcare, and attend at specific times, factors often cited as barriers to engagement [37].

The last two decades have seen a significant shift towards online programs [6], particularly for use with smartphones, however, this has failed to significantly transform rates of retention and engagement, with short-term follow up attrition rates 24% and long-term follow up rates 36%. [7]. Online parenting programs generally offer a modified version of an existing program adapted for an online space, although they still require parents to devote significant time to them, a key barrier to program engagement. As such, even efficacious programs delivered online have high rates of attrition and disengagement [38–40]. Perhaps in part due to these barriers, parents often seek out non-evidence-based online support for immediate answers to their parenting difficulties, making use of parenting forums and social media to ask other parents for advice [41–43], even if parents themselves show scepticism towards some content found online [44]. The results of a qualitative study by Moon et al [45] found that mothers who used the internet for parenting advice felt the amount of advice overwhelming, evidence-based or otherwise. Regardless, parents expressed that a key aspect of online advice was that they were able to find information specific to them. Mothers appreciated the timeliness and relevance of mobile phone apps, allowing access to information when and how they needed it. As such, parenting programs that incorporate content adapted to parent needs, such as support that is relevant and delivered when it is required, may have higher engagement.

Moderate evidence from systematic reviews, interviews, and cross-sectional studies suggests that offering some tailoring within online parenting programs, whereby content is modified to suit individual contexts, improves engagement [42,46–48]. A systematic review of parenting programs (25 studies, combined $N = 5,215$) concluded that tailored parenting programs had higher retention and engagement amongst underserved parents [46]. For example, fathers interviewed in focus groups suggested that they would likely find content related to a specific parenting problem preferential to generic resources [47]. Similarly, a survey of Australian parents found that programs with tailored content were among the most

 

useful when compared to other approaches such as parenting forums or structured online programs [42]. Finan et al [48] also found that parents were more likely to engage with parenting programs when they had some control over program access (i.e., location) and usage (i.e., timing), as employed in adult mental and public health programs via ecological momentary interventions [9,49,50]. These studies indicate that tailoring content in parenting program may lead to improvements in retention and engagement.

Ecological momentary interventions offer potential for providing tailored program content in the moment. These approaches provide treatments or resources to individuals at a time and place most convenient to them [9]. Ecological momentary interventions typically collect 'real-time' responses in order to identify when an individual is experiencing, or is at risk of experiencing, particular symptoms, and then provide strategies to alleviate or prevent those symptoms by immediately delivering a brief intervention [51]. For example, smoking cessation interventions aim to decrease or terminate smoking habits by providing a brief in-the-moment intervention that disrupts individuals' urge to smoke. Interventions can include strategies such as watching videos [52], meditation [52], tips [52,53], and supportive messages [52,53]. A meta-analysis of smoking cessation apps (10 studies, combined $N = 2,391$) concluded that ecological momentary interventions were significantly more likely to promote smoking abstinence compared to controls without an ecological momentary intervention element [54].

It may be possible to apply an ecological momentary design in the context of parenting programs, using real-time data collection to allocate tailored in-the-moment support. Indeed, recent emerging evidence suggests that optimising the timing of parenting programs depending on circumstance improves engagement levels of parents [55]. In a study of 8,074 parents, Cortes et al [55] sent text messages about literacy skills to parents during a weekday or a weekend period, with high-educated parents engaging more with messages scheduled for weekdays and lower-educated parents engaging more with messages scheduled for weekends. The results reinforce that identifying when parents are most receptive to an intervention is important in predicting program efficacy. As in adult mental and public health studies, it may be that parent engagement is highest at times of parent or child emotion dysregulation or difficulty, i.e., when parents are most likely to react punitively or less effectively [56–58]. Alternatively, it is possible that dysregulated moments may not be optimal for offering parents an immediate intervention, as parents may be distracted and focused on managing their child in the moment, either unable to engage with an app or to focus their attention on learning materials. Thus, unlike adult mental and public health programs, it is not clear what the most appropriate timing would be for delivering in-the-moment support to parents.

Ecological momentary assessment (EMA) involves collecting data in the moment multiple times per day [59], and is used to inform the delivery of ecological momentary interventions. EMA is distinct from ecological momentary intervention in that it solely involves the collection of data regarding an individual's current behaviours or thoughts, both as the person experiences them and in the environment that they are experiencing them [59]. Encouragingly, EMAs demonstrate high levels of engagement, with compliance rates above 80% in adult mental and public health contexts [60]. They are therefore ideal to use with programs that rely on accurate, timely data, like smoking cessation programs, which ask whether an individual is craving a cigarette or vape in the moment (i.e., ecological momentary assessment) and, if so, offer an intervention to alleviate the craving (i.e., ecological momentary intervention). Although the use of EMAs in parenting contexts is scarce, two studies have shown promise in improving parent engagement. Dunton et al [61] and Li and Lansford [62] utilised EMA to assess the impact of parent stress on parenting practices, finding that parents completed almost 80% of prompts on average. Dunton et al's [61] study recruited 199 mothers of children aged 8–12 to complete two 7-day periods of EMA data collection, with survey prompts sent up to eight times per day. Li and Lansford [62] recruited 184 parent-child dyads of children with and without ADHD and prompted participants to answer one questionnaire a day for 7 days. Although both studies stated limitations due to participant self-report, both utilised EMA via smartphones, strengthening support for its use in parenting contexts. Conway, Wladis and Hachey [63] highlighted that parents of children aged under 6 years were more time poor than parents of older children, however, both study samples recruited parents of children aged at least 6 years or older on average, which may not reflect the capacity of all parents to maintain high engagement.

These nascent results support the use of EMA to collect in-the-moment responses from some parents, which in turn can be used to inform the development of an ecological momentary intervention as a parenting program.

The current study sought to inform a larger project developing a parenting app by designing a valid and reliable EMA that would be delivered to parents at optimal times of the day to maximise parent engagement. An effective and well-utilised emotion regulation parenting program, underpinned by ecological momentary design, should be (a) brief, thereby reducing burden; (b) valid, thereby accurately measuring dysregulation; and (c) timely, prompting parents when they are most in need of assistance and are most likely to engage. Thus, the current study aims to:

1. Identify items for a very brief EMA which best measures in-the-moment parent and child negative affect and emotion dysregulation;

2. Determine the optimal time of day for participant engagement with smartphone prompts;

3. Assess how often and when specific difficult parenting situations occur; and

4. Establish what times of day and which parenting situations precipitate heightened levels of parent and child negative affect and emotion dysregulation.

## Materials and methods

### Design

The current study used an ecological momentary assessment (EMA) design and involved a series of longitudinal online surveys conducted over a 7-day period. Parents were asked to respond about their oldest child aged 2–4 years. Depending on how many short surveys were completed, parents received a supermarket gift voucher valued at $20, $30, or $50. The study was approved by the Deakin University Human Ethics Advisory Group (HEAG-H 221_2021).

### Recruitment and procedure

Parents were recruited using paid and unpaid methods on social media during March-May 2022. Paid methods used targeted Facebook ads to show parents a post on their News Feed with a link. Unpaid methods involved identifying potentially relevant Facebook groups, requesting permission from administrators to post in the group, and posting a brief study description with a link. On clicking the link for either method, parents were directed to complete a 25-minute Qualtrics baseline questionnaire. Prior to beginning the baseline survey, participants were presented with a Plain Language Statement, after which they were required to provide consent to continue. On completion of the baseline survey, participants were invited to attend an optional onboarding session held via videoconference using the Zoom platform to familiarise them with the EMA platform. Following baseline survey completion, parents were invited to download an app to enrol in the online EMA platform, SEMA3 [64]. The app prompted parents to complete a 1-minute survey five times a day for a one-week period. Parents were evenly allocated to one of two schedules (see S1 Table), with no overlap between schedules to maximise the breadth of hours. Due to uneven attrition between schedule allocation, most parents received schedule 1 ($n=48$).

### Parents

Fig 1 shows the flow of participants through the study. Of the initial 635 baseline responses, 451 parents were excluded primarily due to issues with completeness or eligibility (i.e., not residing in Australia or not a parent of a child aged 2–4 years). A further 28 were excluded due to difficulties enrolling in SEMA3, with a final 47 excluded for not logging into the SEMA3 app and therefore not receiving any EMA surveys. After exclusions, the final sample consisted of 89 parents with full baseline data who engaged with the SEMA3 platform. Three parents included in the final sample received EMA prompts but did not complete any EMA surveys.

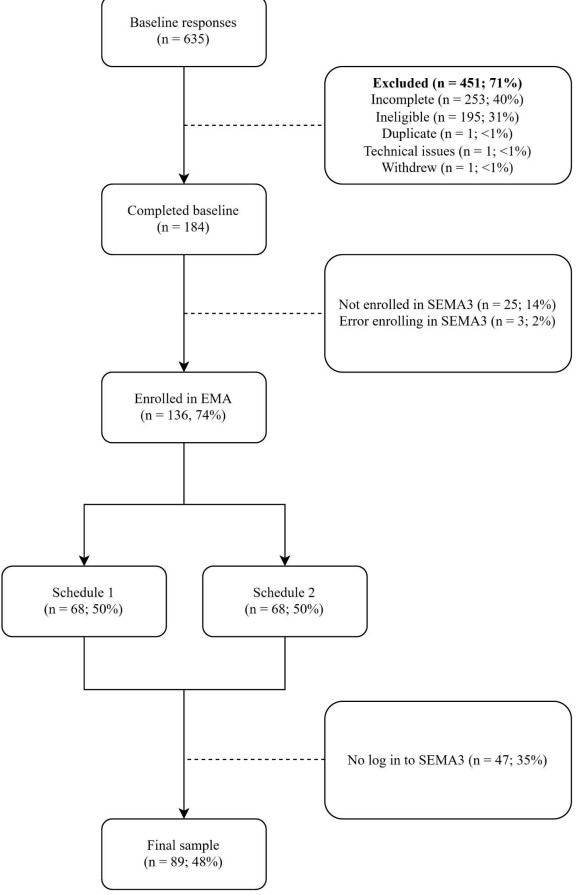

**Fig 1. Flowchart of Participant Retention Through the Study.**

Table 1 summarises the demographic characteristics of the sample. Eligible parents were required to reside in Australia and be a parent of a child aged 2–4 years. Parents were predominantly female, highly educated, and had a medium-high income. Just under half the children were reported as female, and almost half were aged 2 years.

## Measures

In the baseline questionnaire, parents were invited to provide demographic and contact information, as well as information about their parenting, the family emotional climate, and their child's disposition. Although not all measures were validated for use with children aged 2–4 years, measures whose items were deemed appropriate for the age range were selected. On enrolment in SEMA3, parents were asked to complete a short survey about their and their child's in-the-moment emotion dysregulation and negative affect, as well as if they had recently encountered any difficult parenting situations. Based on prior research investigating which situations parents most often solicited advice on parenting forums [41], parents were able to select from a list of 27 pre-populated parenting situations (e.g., 'Child resisting going to bed/sleep'; 'Child fighting with their sibling') or select 'Other' and describe the recent difficult parenting situation (see S2 Table for the full list of parenting situations). These outcomes were selected to align with a larger project aimed at developing a parenting program to improve parent and child emotion regulation. Tables 2 and 3 detail the measures and items used in the baseline questionnaire and short surveys.

**Table 1. Participant demographic characteristics.**

| Baseline characteristic | N (%) |
|---|---|
| Parent age (years), M (SD, range) | 35.6 (4.1, 23-46) |
| 20-29 years old | 2 (2) |
| 30-39 years old | 70 (79) |
| 40-49 years old | 17 (19) |
| Child age (years), M (SD, range) | 2.8 (0.8, 2-4) |
| 2 years old | 40 (45) |
| 3 years old | 26 (29) |
| 4 years old | 23 (26) |
| Parent gender | |
| Female | 76 (85) |
| Male | 12 (13) |
| Non-binary/transgender | 1 (1) |
| Child gender[a] | |
| Female | 43 (48) |
| Male | 44 (49) |
| Single parent | 13 (15) |
| Highest level of parent education | |
| Secondary school not completed | 4 (4) |
| Secondary school completed | 2 (3) |
| Trade certificate/diploma/apprenticeship | 16 (18) |
| Bachelor's degree or above | 67 (75) |
| Employment | |
| Not employed | 25 (28) |
| Part-time | 41 (46) |
| Full-time | 23 (26) |
| Annual household income (AUD) | |
| Up to $52,000 | 13 (15) |
| $52,001-$140,000 | 54 (61) |
| Above $140,000 | 21 (24) |
| Number of children in the home | |
| 1 child | 22 (25) |
| 2 children | 45 (51) |
| 3 children | 11 (12) |
| 4 or more children | 11 (12) |
| Child in early childhood program | |
| Yes | 72 (81) |
| No | 17 (19) |

Note. N = 89.

[a]Two parents responded that their child's gender was female and male twins.

## Data analysis

Data were cleaned and analysed using Stata (Version 16). To address Aim 1, the 19 EMA items assessing parent and child negative affect and emotion dysregulation were evaluated using five purpose-defined criteria: sensitivity, relevance, alignment, frequency, and face validity. As the current study informs a larger project seeking to develop a parenting

**Table 2. Summary of 25-minute baseline questionnaire measures.**

| Construct | Measure |
| --- | --- |
| Child outcomes | |
| Negative affect (2–3 years) | The 12 items used in the Negative Affect scale of the Early Childhood Behavior Questionnaire Very Short Form assessed child negative affect on a 7-point scale (1 = 'Never' to 7 = 'Always', with an 'NA' option). Example item: 'During everyday activities how often does your child seem to be irritated by tags in his/her clothes?'; α = 0.70 [65] |
| Negative affect (4 years) | The 12 items used in the Negative Affect scale of the Childhood Behavior Questionnaire Very Short Form assessed child negative affect on a 7-point scale (1 = 'Extremely true of your child' to 7 = 'Extremely untrue of your child', with an 'NA' option). Example item: 'Gets quite frustrated when prevented from doing something he/she wants to do'; α = 0.75 [66] |
| Depression | The 13-item Short Mood and Feelings Questionnaire (Parent Report) assessed child depressive symptoms on a 3-point scale (1 = 'Not true' to 3 'True'). Example item: 'Your child felt miserable or unhappy'; α = 0.87 [67] |
| Irritability | One item from the CoRonavIruS Health Impact Survey assessed child irritability on a 5-point scale (1 = 'Not irritable or easily angered at all' to 5 'Extremely irritable or easily angered'. Item: 'During the past two weeks, how irritable or easily angered has your child been?' |
| Anxiety | Four items from the Brief Spence Children's Anxiety Scale – Parent Version assessed child anxiety symptoms on a 4-point scale (0 = 'Never' to 3 = 'Always'). Example item: 'Your child worries about things'; α = 0.82 [68] |
| Opposition/Defiance | The eight items used in the Opposition/Defiance Subset of the Swanson, Nolan, and Pelham-IV Questionnaire assessed child opposition on a 4-point scale (0 = 'Not at all' to 3 = 'Very much'). Example item: 'Often loses temper'; α = 0.89 [69] |
| Loneliness | One item from the CoRonavIruS Health Impact Survey assessed child loneliness on a 5-point response scale (1 = 'Not lonely at all' to 5 'Extremely lonely'. Item: 'During the past two weeks, how lonely has your child been?' |
| Temperament (2–3 years) | The eight items used in the Approach and Persistence scales of the revised Toddler Temperament Scale for Children assessed child temperament on a 6-point scale (1 = 'Almost never' to 6 = 'Almost always'). Example item: 'This child is outgoing with adult strangers outside the home'; α = 0.74–0.87 [70] |
| Temperament (4 years) | The eight items used in the Sociability and Persistence scales of the adapted Short Temperament Scale for Children assessed child temperament on a 6-point scale (1 = 'Almost never' to 6 = 'Almost always'). Example item: 'This child is shy with strange adults'; α = 0.83–0.85 [71] |
| Physical health | One item from the Longitudinal Study of Australian Children assessed child physical health on a 5-point scale (1 = 'Excellent' to 5 = 'Poor'). Item: 'In general, your child's current health is...' |
| Parent outcomes | |
| Reflective functioning | The 18 items of the Parental Reflective Functioning Questionnaire assessed parent reflective functioning on a 7-point Likert scale (1 = 'Strongly disagree' to 7 = 'Strongly agree'). Example item: 'The only time I'm certain my child loves me is when he or she is smiling at me'; α = 0.62–0.82 [72] |
| Beliefs about children's emotions | 26 items from the Parents' Beliefs about Children's Emotions questionnaire assessed parent beliefs about children's emotions on a 6-point scale (1 = 'Strongly disagree' to 6 = 'Strongly agree'). Example item: 'Children use emotions to manipulate others'; α = 0.60–0.71 [73] |
| Emotional expression | The 24 items of the Self-Expressiveness in the Family Questionnaire assessed family emotional expressiveness on a 9-point scale (1 = 'Not at all frequently in my family' to 9 = 'Very frequently in my family'). Example item: 'Showing contempt for another's actions'; α = 0.82–0.88 [74] |
| Emotion regulation | The 16 items of the Difficulties in Emotion Regulation Scale – 16 item version and three items from the Difficulties in Emotion Regulation Scale assessed parent emotion regulation on a 5-point scale (1 = 'Almost never' to 5 = 'Almost always'). Example item: 'I have difficulty making sense out of my feelings'; α = 0.92–0.94 [75] |
| Family environment | |
| Reading in the home | One item from the home literacy environment measure assessed parent-to-child reading frequency on a 4-point scale (0 = 'Not at all' to 3 'Every day'). Item: 'In a typical week, how often do you read books to your child? |
| Books in the home | One item from the home literacy environment measure assessed the number of children's books in the home on a 3-point scale (0 = 'Less than 10' to 2 'More than 30'. Item: 'Approximately how many books does your child own?' |
| Psychological distress | The six items of the Kessler-6 assessed parent distress on a 5-point scale (0 = 'None of the time' to 4 = 'All of the time'). Example item: 'Thinking about yourself in the past four weeks, about how often did you feel nervous?'; α = 0.89 [76] |
| Positive affect | The five items used in the Positive Affect subscale of the International Positive and Negative Affect Schedule – Short Form assessed parent positive affect on a 5-point scale (1 = 'Very slightly or not at all' to 5 = 'Extremely'). Example item: 'Thinking about yourself and how you normally feel, to what extent do you generally feel alert?'; α = 0.80 [77] |

*(Continued)*

Table 2. (Continued)

| Construct | Measure |
|---|---|
| Stress | The seven items used in the Stress scale of the Depression Anxiety Stress Scales-21 assessed parent stress on a 4-point scale (0 = 'Did not apply to me at all' to 3 = 'Applied to me very much or most of the time'). Example item: 'I found it hard to wind down'; α = 0.91 [78] |
| Life event stress | The eight items adapted from the List of Threatening Experiences Questionnaire assessed parent significant events in the past 12 months on a yes/no scale. Example item: 'You became pregnant or had a baby'; κ = 0.72 [79] |
| Social support | One item from the Longitudinal Study of Australian Children assessed parent social support on a 3-point scale (1 = 'I get enough help/I don't need any help' to 3 = 'I don't get any help at all'). Item: 'Overall, how do you feel about the amount of support or help you get from family or friends living elsewhere?' |
| Verbal inter-parental conflict | The four items adapted from the Quality of Co-Parental Communication scale assessed verbal inter-parental conflict on a 5-point scale (1 = 'Never' to 5 = 'Always'). Example item: 'How often do you and your partner disagree about basic child-rearing issues?'; α = 0.88–0.89 [80] |
| Physical inter-parental conflict | One item from the Longitudinal Study of Australian Children assessed physical inter-parental conflict 5-point scale (1 = 'Never' to 5 = 'Always'). Item: 'How often do you have arguments with your partner that end up with people pushing, hitting, kicking or shoving?' |
| Demographics | |
| Parent and family characteristics | Parent age; parent gender; parent country of birth; parent country of residence; parent postcode; parent/child relationship; parent relationship status; language spoken at home; children in the household; parent schooling; parent qualification; parent study status, parent employment status; parent/partner living arrangement; partner/child relationship; partner employment status; household income |
| Child characteristics | Child age; child gender, child living arrangement; child education program |

**Table 3. Summary of 1-minute survey measures.**

| Construct | Measure |
|---|---|
| Parent outcomes | |
| Emotion dysregulation | The five items used in the State Difficulties in Emotion Regulation Scale assessed in-the-moment parent emotion dysregulation on an 11-point scale (0 = 'Not at all' to 10 = 'Extremely'). Example item: 'Indicate how much each statement applies to your emotions right now: My emotions feel overwhelming'; α = 0.86 [81] |
| Negative affect | The five items used in the Negative Affect scale of the international Positive and Negative Affect Schedule Short Form assessed parent negative affect on an 11-point scale (0 = 'Never' to 10 = 'Always'). Example item: 'Indicate to what extent you feel this way right now: Upset'; α = 0.76 [77] |
| Child outcomes | |
| Emotion dysregulation | The four items adapted from the State Difficulties in Emotion Regulation Scale assessed in-the-moment child emotion dysregulation on an 11-point scale (0 = 'Not at all' to 10 = 'Extremely'). Example item: 'Indicate how much each statement applies to your child's emotions right now: My child seems to be overwhelmed by their emotions' [81] |
| Negative affect | The five items adapted from the Negative Affect scale of the Positive and Negative Affect Schedule for Children – Parent Shortened Version assessed child negative affect on an 11-point scale (0 = 'Never' to 10 = 'Always'). Example item: 'Indicate to what extent your child feels this way right now: Depressed'; α = 0.83 [82] |
| Difficult parenting situation | One item assessed recent difficult parenting situations. Item: 'Which of the following situations have you struggled to manage with your child most recently?' |

program as a smartphone app, these criteria were decided as being relevant to the overall goals. First, sensitivity was defined as how well an individual EMA item was associated with the composite total of EMA items for the given measure. To assess sensitivity, a series of multilevel mixed-effects linear regression models were conducted with the relevant EMA composite measure as the dependent variable and the individual EMA item as the independent variable. A higher unstandardised regression coefficient was considered a higher level of sensitivity.

Second, relevance was defined as how well an individual EMA item was associated with a set of 26 baseline constructs measuring child outcomes (negative affect, depressive symptoms, anxiety symptoms, oppositional-defiance behaviour

problems, and temperament), parent outcomes (parental reflective functioning, beliefs about children's emotions, emotion regulation, psychological distress, negative affect, and stress), and family functioning (expressiveness in the family and inter-parental conflict). To assess relevance, a series of multilevel mixed-effects linear regression models were conducted with individual EMA items as the dependent variable and each baseline composite measure as the independent variable. A new count variable was derived, representing the overall number of significant associations evident between EMA items and baseline measures, with a higher number of significant associations equating to higher relevance.

Third, alignment was defined as how well an individual EMA item was associated with other EMA items that were not part of same measure, based on comparing the mean unstandardised regression coefficients from a series of multilevel mixed-effects linear regression analyses. The item of interest was the dependent variable and the non-parent measure item was the independent variable. A higher mean coefficient indicated stronger alignment for an item.

Fourth, frequency was assessed as a count variable, capturing the number of parents who never reported parent/child negative affect or emotion dysregulation on a given EMA item (i.e., never responding above 0 'Not at all') across the week. A higher frequency score represented a greater number of parents who never reported symptoms on a given EMA item.

Fifth, face validity was defined in terms of the appropriateness of each EMA item in assessing parent and child change associated with delivery of an emotion-focused parenting app for parents of children aged 2–4 years. After items were assessed using the first four criteria, the research team discussed each item and its appropriateness in context of the targeted behaviour within the parenting program.

Finally, each item was ranked 1–5, with a lower number indicating a higher preference from the research team. Results from the five criteria were used to identify which of the 19 short survey items were best to use for further analyses.

To address Aim 2, the optimal time for participant engagement was determined using two metrics: the time of day in which parents responded to smartphone prompts and the time of day that parents were with their child. Response rate was assessed by examining the time of day when a participant missed survey prompts, whereas child presence was based on responses to a specific short survey item. Participant engagement was considered higher when a participant both engaged with a prompt and was with their child.

To address Aim 3, the occurrence of specific difficult parenting situations was assessed by the number of occasions in which parents selected a situation as the most recently occurring, with a higher response rate for a situation indicating a more common occurrence, as well as the time of day in which they occurred.

To address Aim 4, mean item scores from items selected in Aim 1 were compared across the 20 most commonly selected difficult parenting situations and across different times of the day, with higher scores indicating higher levels of parent and child negative affect or emotion dysregulation.

## Results

### Item selection

Table 4 presents an overview of results according to the five criteria (sensitivity; relevance; alignment; frequency; face validity) used to evaluate the 19 short survey items. Six items were selected for inclusion in the subsequent analyses, based on the ranking assigned to each item within each measure, with the final selection including one item measuring parent negative affect, two measuring parent emotion dysregulation, two measuring child negative affect, and one measuring child emotion dysregulation.

For parent negative affect, item 1 (Upset) was selected as most appropriate. Item 1 had the strongest alignment and highest frequency. In terms of face validity, the item was considered as having flexibility in reflecting different types of parent affect. Items 3 (Ashamed) and 5 (Afraid) were highest for sensitivity, however, were not selected as more than 30% of parents reported never experiencing those emotions during the one-week study period. Item 2 (Hostile) was not selected as the research team decided it was not an appropriate emotion to measure parent negative affect, whilst item 4 (Nervous) was discounted in favour of item 1.

**Table 4. Evaluation of short survey items' sensitivity, relevance, alignment, frequency, and validity.**

| Item | | Sensitivity[a] | Relevance[b] | Alignment[c] | Frequency[d] | Validity[e] | Ranking[f] |
|---|---|---|---|---|---|---|---|
| Parent negative affect | | | | | | | |
| 1 | **Upset** | **2.13** | **8** | **0.42** | **9** | **1** | **1** |
| 2 | Hostile | 2.50 | 5 | 0.24 | 22 | 2 | 3 |
| 3 | Ashamed | 2.81 | 8 | 0.19 | 27 | 2 | 4 |
| 4 | Nervous | 1.92 | 15 | 0.22 | 11 | 2 | 2 |
| 5 | Afraid | 2.90 | 11 | 0.10 | 38 | 2 | 5 |
| Child negative affect | | | | | | | |
| 1 | Depressed | 3.80 | 8 | 0.08 | 62 | 3 | 3 |
| 2 | **Angry** | **1.88** | **8** | **0.31** | **19** | **1** | **1** |
| 3 | Scared | 2.96 | 10 | 0.19 | 39 | 3 | 4 |
| 4 | Afraid | 3.11 | 13 | 0.15 | 49 | 3 | 5 |
| 5 | **Sad** | **1.86** | **3** | **0.38** | **14** | **2** | **2** |
| Parent emotion dysregulation | | | | | | | |
| 1 | **My emotions feel overwhelming** | **2.11** | **13** | **0.44** | **6** | **1** | **1** |
| 2 | **I am having difficulty controlling my behaviours** | **2.66** | **13** | **0.30** | **18** | **2** | **2** |
| 3 | I am having difficulty doing things I need to do right now | 1.72 | 13 | 0.41 | 5 | 2 | 3 |
| 4 | I am paying attention to how I feel | 0.85 | 2 | 0.15 | 0 | 3 | 5 |
| 5 | I have no idea how I am feeling | 1.63 | 4 | 0.13 | 16 | 3 | 4 |
| Child emotion dysregulation | | | | | | | |
| 1 | My child seems to be overwhelmed by their emotions | 3.13 | 9 | 0.49 | 16 | 2 | 3 |
| 2 | My child seems to be feeling out of control | 3.68 | 8 | 0.42 | 19 | 2 | 2 |
| 3 | **My child has difficulty controlling their behaviours** | **3.19** | **6** | **0.49** | **8** | **1** | **1** |
| 4 | My child has difficulty doing things they need to do now | 2.58 | 10 | 0.54 | 8 | 3 | 4 |

Items in bold were final items selected as best measuring parent and child negative affect and emotion dysregulation.

[a]Results are unstandardised regression coefficients.

[b]Number of significant associations with baseline measures ($p < 0.05$).

[c]Results are means of unstandardised regression coefficients.

[d]86 parents had short survey data.

[e]The research team ranked more face valid items lower.

[f]Lower scores indicate a stronger overall team ranking for the item.

For child negative affect, items 2 (Angry) and 5 (Sad) were selected as most appropriate. Items 2 and 5 had second highest and highest alignment, respectively, whilst the research team also decided that it was important to capture emotions that indicated parents and children were most at risk of emotion dysregulation. Additionally, items 1 (Depressed), 3 (Scared), and 4 (Afraid) were never reported as the child's in-the-moment emotion by more than 45% of parents, which led to those three items being discounted.

For parent emotion dysregulation, items 1 ('My emotions feel overwhelming') and 2 ('I am having difficulty controlling my behaviours') were selected as most appropriate. Item 1 had the strongest alignment and equal-highest relevance, with strong face validity. The team considered item 1's mention of emotions to align with the objective of an emotion regulation parenting app, and item 2 had the highest sensitivity and equal-highest relevance. Item 3 ('I am having difficulty doing the things I need to do right now') had equal-highest relevance and second-highest alignment, however, the research team considered that participant responses to the item may be related to factors not relevant to emotion regulation (e.g., child behaviour, other contextual factors), and it was therefore discounted. Items 4 ('I am paying attention to how I feel') and 5 ('I have no idea how I am feeling') scored poorly across almost all criteria and were therefore not considered for selection.

For child emotion dysregulation, item 3 ('My child has difficulty controlling their behaviours') was selected as most appropriate. Item 3 had the equal-highest frequency, and second highest and equal-second highest sensitivity and alignment. In terms of face validity, the research team considered item 3 might detect child emotional distress which parents perceived solely as behavioural problems in the moment. Item 4 ('My child has difficulty doing things they need to do now') had the highest relevance and alignment, however, the research team decided that the item was too broad and may not be relevant for children aged 2–4 years, and as such it was discounted. Item 1 ('My child seems to be overwhelmed by their emotions') had the second highest relevance and equal-second-highest alignment, and item 2 ('My child seems to be feeling out of control') had the highest sensitivity, however, parents less often reported the statements as being reflective of their child in the moment than item 3 (i.e., frequency), leading to both items being less preferred than item 3.

### Survey engagement

**Smartphone prompt response rate.** Table 5 shows when parents were most likely to respond to smartphone prompts. Parents were less likely to respond to smartphone prompts during the early morning, the start of the workday, and at night, with no major discrepancies in response rates at other times of the day.

**Frequency of parent and child together.** Table 6 shows whether parents reported that they were with their children while responding to a smartphone prompt. Children were most often with their parents during the morning, specifically 7:30am, and early evening (6:00 pm). In contrast, parents were least likely to be with their children at night, as well as during working hours.

**Timing of difficult parenting situations.** Table 7 reports the 20 parenting situations (of 27 tested) rated as the most common by parents, and the times at which they were most often reported. Ratings for some situations were influenced by the time of day. For example, a child resisting going to bed was reported as a difficult parenting situation almost 70% of the time from 7:30 pm onwards, whereas a child whining for more screen time did not fluctuate significantly throughout the day.

**Heightened parent and child negative affect and emotion dysregulation.** Table 8 reports mean scores for parent and child negative affect and emotion dysregulation associated with the 20 most common difficult parenting situations. Parents reported the highest levels of parent negative affect when their child refused to hold their hand whilst crossing the road and highest emotion dysregulation when their child resisted going to bed. Meanwhile, parents reported that their child showed highest levels of negative affect when the child did not want to leave the parent's side and highest

**Table 5. Frequency of participant responses to smartphone prompts during the day.**

| Time of day | Surveys completed (*n*, %) |
| --- | --- |
| 6:30am | 232 (75.32) |
| 7:30am | 218 (83.21) |
| 9:00am | 235 (77.56) |
| 11:00am | 206 (81.10) |
| 1:00 pm | 269 (82.52) |
| 2:30 pm | 207 (83.47) |
| 4:30 pm | 271 (83.13) |
| 6:00 pm | 219 (81.72) |
| 7:30 pm | 263 (85.11) |
| 9:00 pm | 209 (77.41) |

Response frequency for each hour differs due to differences in schedule attrition rates.

**Table 6. Frequency of the times of day parents are with their child.**

| Time of day | Parent with their child (*n*, %) |
|---|---|
| 6:30am | 142 (62.28) |
| 7:30am | 159 (73.61) |
| 9:00am | 142 (61.21) |
| 11:00am | 115 (56.93) |
| 1:00 pm | 145 (54.31) |
| 2:30 pm | 97 (47.78) |
| 4:30 pm | 184 (68.91) |
| 6:00 pm | 174 (81.31) |
| 7:30 pm | 144 (55.17) |
| 9:00 pm | 81 (39.13) |

**Table 7. Most common parenting situations and their frequency during the day.**

| Parenting situation (frequency) | 6:30am | 7:30am | 9:00am | 11:00am | 1:00 pm | 2:30 pm | 4:30 pm | 6:00 pm | 7:30 pm | 9:00 pm |
|---|---|---|---|---|---|---|---|---|---|---|
| | *n (%)* | *n (%)* | *n (%)* | *n (%)* | *n (%)* | *n (%)* | *n (%)* | *n (%)* | *n (%)* | *n (%)* |
| Being fussy about food (173) | 11 (6) | 17 (10) | 19 (11) | 14 (8) | 15 (9) | 11 (6) | 16 (9) | **36 (21)** | 19 (11) | 15 (9) |
| Resisting going to bed/sleep (166) | 11 (6) | 4 (2) | 3 (2) | 1 (1) | 13 (8) | 6 (3) | 6 (3) | 10 (6) | **56 (34)** | **56 (34)** |
| Fighting with their sibling (130) | 18 (14) | 12 (9) | 15 (12) | 10 (8) | 13 (10) | 3 (2) | **19 (15)** | 13 (10) | 15 (12) | 12 (9) |
| Refusing to stop activity (114) | 3 (3) | 10 (9) | 12 (11) | 12 (11) | **26 (23)** | 9 (8) | 19 (17) | 10 (9) | 4 (4) | 9 (8) |
| Resisting getting dressed (110) | 11 (10) | 22 (20) | **33 (30)** | 4 (4) | 7 (6) | 5 (5) | 7 (6) | 7 (6) | 11 (10) | 3 (3) |
| Whining for more screen time (80) | 6 (8) | 7 (9) | 10 (13) | 8 (10) | 7 (8) | 9 (11) | **14 (18)** | 9 (11) | 8 (10) | 2 (3) |
| Struggling to sit for mealtime (76) | 3 (4) | 11 (14) | 8 (11) | 4 (5) | 8 (11) | 7 (9) | 3 (4) | 13 (17) | **17 (22)** | 2 (3) |
| Not staying in bed (72) | **20 (28)** | 10 (14) | 1 (1) | 5 (7) | 1 (1) | 3 (4) | 4 (6) | 0 (0) | 11 (15) | 17 (24) |
| Not leaving parent/refusing to play (57) | 3 (5) | 9 (16) | 11 (19) | **12 (21)** | 4 (7) | **12 (21)** | 2 (4) | 3 (5) | 0 (0) | 1 (2) |
| Whining for a snack (53) | 3 (6) | 1 (2) | 7 (13) | 6 (11) | 4 (8) | 7 (13) | 9 (17) | **12 (23)** | 4 (8) | 0 (0) |
| Resisting a bath/shower (43) | 2 (5) | 3 (7) | 0 (0) | 3 (7) | 0 (0) | 1 (2) | 5 (12) | 10 (23) | **13 (30)** | 6 (14) |
| Not putting shoes on (42) | 2 (5) | 4 (10) | 6 (14) | 10 (24) | 3 (7) | 5 (12) | 8 (19) | 2 (5) | 0 (0) | 2 (5) |
| Refusing to play with their sibling (40) | 2 (5) | 3 (8) | 0 (0) | **7 (18)** | 6 (15) | **7 (18)** | 4 (10) | 5 (13) | 3 (8) | 3 (8) |
| Meltdown when screen turned off (39) | 2 (5) | 5 (13) | 3 (8) | 4 (10) | 4 (10) | 5 (13) | **7 (18)** | 1 (3) | 4 (10) | 4 (10) |
| Resisting brushing their teeth (29) | 2 (7) | 3 (10) | 0 (0) | 0 (0) | 0 (0) | 0 (0) | 1 (3) | 4 (14) | **12 (41)** | 7 (24) |
| Resisting getting in/out of the car (26) | 1 (4) | 1 (4) | 2 (8) | 1 (4) | 5 (19) | 2 (8) | **8 (31)** | 4 (15) | 2 (8) | 0 (0) |
| Complaining of hunger/thirst (20) | 2 (10) | 2 (10) | 0 (0) | 1 (5) | 1 (5) | 1 (5) | 0 (0) | 4 (20) | 4 (20) | **5 (25)** |
| Insisting on inappropriate outfit (16) | 2 (13) | **6 (38)** | 1 (6) | 1 (6) | 1 (6) | 2 (13) | 1 (6) | 1 (6) | 0 (0) | 1 (6) |
| Not accepting help to dress (13) | 2 (15) | 0 (0) | **5 (38)** | 1 (8) | 3 (23) | 0 (0) | 0 (0) | 0 (0) | 1 (8) | 1 (8) |
| Struggling to leave the other parent (13) | 1 (8) | 1 (8) | **1 (23)** | 1 (8) | 1 (15) | 0 (0) | 1 (8) | **1 (23)** | 1 (8) | 0 (0) |
| Not holding hands whilst crossing (13) | 0 (0) | 1 (8) | 1 (8) | 0 (0) | **4 (31)** | 3 (23) | 2 (15) | 2 (15) | 0 (0) | 0 (0) |

Items in bold are the time during which each difficult parenting situation was most often selected.

emotion dysregulation both when their child did not want to leave the parent's side and when the child refused to put their shoes on.

Table 9 shows the mean scores of parent and child negative affect and emotion dysregulation throughout the day. Parents reported heightened levels of negative affect and emotion dysregulation in the evening, whereas child negative affect was higher in the late afternoon and emotion dysregulation was higher in the evening.

**Table 8. Most common parenting situations with means and 95% confidence intervals of parent and child negative affect and emotion dysregulation.**

| Parenting situation (frequency) | Parent outcomes, M (95% CI (LL, UL)) | | Child outcomes, M (95% CI (LL, UL)) | |
|---|---|---|---|---|
| | Negative affect | Emotion dysregulation[a] | Negative affect[a] | Emotion dysregulation |
| Being fussy about food (173) | 0.83 (0.56, 1.10) | 1.03 (0.75, 1.32) | 0.70 (0.50, 0.89) | 1.42 (1.10, 1.74) |
| Resisting going to bed/sleep (166) | 1.20 (0.87, 1.53) | **1.21 (0.89, 1.52)** | 1.08 (0.70, 1.46) | 1.45 (0.99, 1.91) |
| Fighting with their sibling (130) | 0.85 (0.60, 1.10) | 0.88 (0.62, 1.13) | 0.63 (0.36, 0.91) | 1.04 (0.71, 1.37) |
| Refusing to stop activity (114) | 0.89 (0.58, 1.20) | 0.88 (0.59, 1.18) | 1.02 (0.64, 1.40) | 1.55 (1.05, 2.04) |
| Resisting getting dressed (110) | 0.70 (0.44, 0.96) | 0.86 (0.60, 1.13) | 0.67 (0.37, 0.96) | 1.17 (0.75, 1.59) |
| Whining for more screen time (80) | 0.88 (0.48, 1.28) | 0.90 (0.50, 1.30) | 0.45 (0.18, 0.71) | 0.92 (0.45, 1.40) |
| Struggling to sit for mealtime (76) | 0.88 (0.48, 1.28) | 0.90 (0.48, 1.31) | 0.75 (0.41, 1.08) | 1.54 (1.05, 2.04) |
| Not staying in bed (72) | 0.83 (0.44, 1.22) | 0.85 (0.49, 1.20) | 0.67 (0.33, 1.00) | 1.14 (0.56, 1.72) |
| Not leaving parent/refusing to play (57) | 0.41 (0.15, 0.67) | 0.72 (0.37, 1.08) | **1.13 (0.16, 2.09)** | **1.63 (0.35, 2.90)** |
| Whining for a snack (53) | 1.08 (0.66, 1.49) | 1.16 (0.69, 1.63) | 0.67 (0.36, 0.98) | 1.19 (0.63, 1.75) |
| Resisting a bath/shower (43) | 0.38 (0.24, 1.49) | 0.62 (0.20, 1.04) | 0.40 (0.07, 0.73) | 1.08 (0.32, 1.85) |
| Not putting shoes on (42) | 1.00 (0.28, 1.72) | 0.78 (0.40, 1.15) | 0.60 (0.20, 1.00) | **1.63 (0.72, 2.53)** |
| Refusing to play with their sibling (40) | 0.75 (0.28, 1.22) | 0.81 (0.39, 1.23) | 0.93 (0.39, 1.48) | 1.48 (0.95, 2.02) |
| Meltdown when screen turned off (39) | 0.82 (0.23, 1.40) | 0.57 (0.22, 0.91) | 0.55 (0.03, 1.06) | 1.39 (0.47, 2.31) |
| Resisting brushing their teeth (29) | 0.62 (0.17, 1.07) | 1.14 (0.43, 1.85) | 0.30 (0.00, 0.69) | 1.00 (0.31, 1.69) |
| Resisting getting in/out of the car (26) | 0.23 (0.00, 0.52) | 0.27 (0.03, 0.51) | 0.19 (0.00, 0.42) | 0.39 (0.00, 0.78) |
| Complaining of hunger/thirst (20) | 1.45 (0.24, 2.66) | 0.76 (0.19, 1.33) | 0.23 (0.00, 0.58) | 0.30 (0.00, 0.78) |
| Insisting on inappropriate outfit (16) | 0.13 (0.00, 0.39) | 0.07 (0.00, 0.20) | 0.50 (0.14, 0.86) | 0.80 (0.13, 1.47) |
| Not accepting help to dress (13) | 0.23 (0.00, 0.50) | 0.35 (0.00. 0.74) | 0.67 (0.03, 1.30) | 0.83 (0.04, 1.62) |
| Struggling to leave the other parent (13) | 0.17 (0.00, 0.53) | 0.85 (0.12, 1.57) | 0.50 (0.00, 1.58) | 0.40 (0.00, 1.51) |
| Not holding hands whilst crossing (13) | **1.53 (0.29, 2.79)** | 1.13 (0.35, 1.90) | 0.71 (0.00, 1.71) | 1.17 (0.04, 2.37) |

Items in bold were the highest mean score for each outcome.

[a]Mean score is the average of the two items selected for the outcome.

**Table 9. Comparison of times of day with means and 95% confidence intervals of parent and child negative affect and emotion dysregulation.**

| Time of day | Parent measures, M (95% CI (LL, UL); n) | | Child measures, M (95% CI (LL, UL); n) | |
|---|---|---|---|---|
| | Negative affect | Emotion dysregulation[a] | Negative affect[a] | Emotion dysregulation |
| 6:30am | 0.65 (0.48, 0.82; 230) | 0.61 (0.46, 0.75; 227) | 0.47 (0.32, 0.62; 141) | 0.73 (0.52, 0.94; 140) |
| 7:30am | 0.85 (0.61, 1.08; 214) | 0.83 (0.59, 1.06; 209) | 0.56 (0.40, 0.73; 150) | 1.10 (0.82, 1.38; 153) |
| 9:00am | 0.74 (0.54, 0.94; 232) | 0.76 (0.58, 0.98; 230) | 0.70 (0.50, 0.91; 140) | 1.04 (0.74, 1.33; 140) |
| 11:00am | 0.82 (0.58, 1.06; 201) | 0.91 (0.67, 1.14; 197) | 0.63 (0.39, 0.86; 108) | 0.98 (0.67, 1.30; 110) |
| 1:00 pm | 0.74 (0.55, 0.92; 266) | 0.68 (0.53, 0.84; 263) | 0.84 (0.56, 1.11; 145) | 1.31 (0.95, 1.67; 144) |
| 2:30 pm | 0.61 (0.40, 0.83; 199) | 0.63 (0.46, 0.80; 188) | 0.66 (0.38, 0.94; 88) | 1.21 (0.78, 1.64; 90) |
| 4:30 pm | 0.74 (0.54, 0.93; 267) | 0.69 (0.54, 0.85; 264) | **0.85 (0.64, 1.07; 177)** | 1.31 (1.00, 1.62; 180) |
| 6:00 pm | 0.80 (0.56, 1.05; 210) | **0.99 (0.74, 1.23; 201)** | 0.63 (0.45, 0.80; 168) | 1.28 (1.00, 1.57; 165) |
| 7:30 pm | **1.00 (0.80, 1.20; 262)** | 0.86 (0.68, 1.04; 260) | 0.82 (0.57, 1.07; 141) | **1.71 (1.34, 2.08; 140)** |
| 9:00 pm | 0.63 (0.41, 0.86; 202) | 0.81 (0.58, 1.05; 194) | 0.64 (0.36, 0.91; 77) | 0.61 (0.33, 0.90; 75) |

Items in bold were the highest mean score for each outcome.

[a]Mean score is the average of the two items selected for the outcome.

## PLOS One

## Discussion

Findings from the current study will inform the development of an in-the-moment parenting smartphone app to improve emotion regulation in children aged 2–4 years. The study used an EMA approach to collect information from 89 parents with a child aged 2–4 years to determine the (1) optimal item selection for measuring in-the-moment parent and child negative affect and emotion dysregulation, (2) timing of prompts, (3) selection of daily experiences of difficult parenting situations in an Australian setting, as well as to (4) establish times of day and parenting situations that precipitate heightened levels of parent and child negative affect and emotion dysregulation.

For aim 1, six items of 19 tested were selected to measure parent and child negative affect and emotion dysregulation: 'Upset' (parent negative affect); 'Angry' and 'Sad' (child negative affect); 'My emotions feel overwhelming' and 'I am having difficulty controlling my behaviours' (parent emotion dysregulation); and 'My child has difficult controlling their behaviours' (child emotion dysregulation). No specific recommendations currently exist to guide the development of an EMA measure, but previous research using brief EMA measures utilises a wide range of approaches to assess reliability and validity [83,84]. Thus, our study extends the EMA literature by providing a coherent and detailed assessment framework including five elements: sensitivity, relevance, alignment, frequency, and face validity. While consideration was given to the psychometric strengths of each item (sensitivity, relevance, alignment, and frequency), the face validity of each item was given the greatest weight due to the lack of existing parenting EMA studies. This approach of prioritising face validity in item selection for EMA measures has been utilised by other studies that sought to develop very brief measures [83,85].

Aim 2 of the current study sought to identify when parents would be most likely to respond to smartphone prompts and to be with their children. Parents were least likely to respond to prompts at 6:30am, 9:00am, and 9:00 pm. While parents were not asked why they were unable to complete prompts, it may be that families were not awake or busy with children at 6:30am. Parents may be beginning their workday at 9:00am, and at 9.00 pm might be tired from the day or avoiding technology. In general, parents reported most often being with their child at 7:30am and 6:00 pm, however, this varied greatly between weekdays and weekends – on weekends, parents reported being with their child 10−30% more often during the hours of 9:00am-4:30 pm. As more than 70% of the sample was employed, and 80% of the children in the sample were enrolled in early childhood programs, parents were likely to be at work during the hours of 9:00am-4:30 pm on a weekday, rather than with their child. Parents may have indicated that they were not with their child at 6:30am and 9:00 pm because their child was asleep, even if they were in the same house. Results showed the average participant response rate was 81%, similar to other studies using EMA in parenting contexts, [61,62], and EMA more generally [34]. Although the result should be interpreted with caution as participants were only involved for one week and were paid to participate, this response rate is surprising considering previous evidence that suggests parents with children under 6 years are more time poor compared to other parent groups [63]. Overall, results support the use of EMA to improve parent engagement and suggest the need for tailoring the timing of EMA prompts to maximise the opportunity for parents to respond.

Aim 3 investigated the most common difficult parenting situations and times of day when they occurred. The five most common situations included difficulties with food, bedtime, sibling conflict, transitions, and screens, however, the time of day in which they occurred varied by situation. Situations related to routines or daily rhythms were more often reported in the morning or evening, while other situations were reported evenly throughout the day. For instance, children fighting with their sibling was reported relatively evenly across the course of the day, while children resisting going to bed was typically reported in the two latest time periods (7:30 pm and 9:00 pm). Parenting programs often use a one-size-fits-all approach to content, without consideration for the unique circumstances of families [42,47]. Such an approach results in parents disengaging from programs, as they do not feel that the resources provided reflect their individual needs as parents and families.

Aim 4 of the current study sought to identify the types of parenting situations and times of day that led to heightened parent and child negative affect and emotion dysregulation. In general, there was no pattern of dysregulation for parents

or children when it came to parenting situations. Some situations that were reported infrequently resulted in high levels of dysregulation, while other more frequent situations did not elicit a similar response. For example, parents responded that it was rare for children to resist holding their hand when they crossed the road, however, on the occasions that it did happen, parents showed higher levels of negative affect compared to other situations. Conversely, when parents reported that their child was fighting with their siblings, they did not report higher dysregulation in either themselves or their child compared to other situations. It may be that parents and children naturally learn coping strategies for situations which are more common, thereby reducing in-the-moment dysregulation. Further to this, parents and children did not necessarily show heightened regulation in the same situations. Children appeared to exhibit highest levels of negative affect and emotion dysregulation in situations where they needed to leave their parents, whereas parents reported highest negative affect when their child did not hold their hand whilst crossing the road. These discrepancies could be due to parents' and children's different perceptions of danger, with parents more fearful of the immediate danger or fast-moving traffic and children more uneasy about being away from their parents.

Compared to parenting situations, time of day appeared to show a more consistent pattern whereby both parents and children showed higher dysregulation from the late afternoon onwards and seemingly peaking at around 7:30 pm. This suggests that coping for both parents and children is lower later in the day. This may be because all parties have fewer emotional resources after work or childcare, which may then be exacerbated by evening routines that cause conflict. This is emphasised by the fact that children resisting going to bed was the second-most reported parenting situation, which also resulted in high levels of negative affect and emotion dysregulation for all. The final reported time of day, at 9:00 pm, showed a noticeable decrease in dysregulation, potentially suggesting that once children were asleep, parents were able to report feeling less dysregulated. The results reinforce that the development of parenting programs would benefit from consideration of the contexts in which parents request an intervention and the times at which they are most likely to require support.

Adult mental and public health programs have had success in offering ecological momentary interventions that provide contextual, in-the-moment resources. Our findings offer concrete information on how similar designs could be implemented within a mental health prevention context, such as offering parenting support via ecological momentary interventions to young families. Our study was specifically designed to inform the development of a parenting program that aims to improve parent and child emotion regulation using an ecological momentary design, although the findings are likely to have relevance to any program using a similar design. EMAs provide in-the-moment data that reflect an individual's experience of their environment in real time, allowing for parents to provide more accurate responses about their and their child's affect and emotions. Evidence suggests that parents prefer programs with content tailored to their context [46], and the diversity in the types of difficult situations that parents reported and the times during which they occurred throughout the current study indicates that tailoring content to reflect all kinds of family situations is warranted when developing future parenting programs.

In considering whether to use EMA for parenting programs, the current study examined whether a smartphone-based approach would result in higher levels of engagement by parents and how programs could maximise the efficacy of their content. Our findings offer concrete information on how similar designs could be implemented within a mental health prevention context. In understanding the barriers that families face with in-person programs, online approaches can help to mitigate some of these obstacles whilst reaching families that are most substantially limited by them. In turn, with parents indicating their desire for content that more accurately reflects their own personal circumstances, providing opportunities for support and learning both when parents can access them and in times when their needed and convenient may precipitate stronger engagement and improved outcomes. Parents expressed a desire for content that accurately represents their circumstances, with flexible access to support and learning opportunities. Our findings align with evidence from Finan et al [48], Hansen, Broomfield and Yap [46], and Sicouri et al [47], indicating that parents prefer programs tailored to their context. The diversity in reported parenting challenges – both in type and timing – suggests that future programs should

reflect a wide range of family situations. Ensuring availability when needed and at convenient times may enhance engagement and improve outcomes.

## Strengths and limitations

A strength of the current study was its use of an EMA design to capture participant data, which has previously been shown to result in high levels of engagement [34,61,62]. Of the 89 parents in the final sample, 61 completed at least 30 of a possible 35 short surveys, with 81% of all surveys completed, supporting the feasibility of EMAs as an engaging approach to parenting interventions. Furthermore, the current study captured both a wide range of parenting situations and a broad period of time, providing data on a diverse set of circumstances which reflect differences both within and between families' daily lives.

The study's findings, however, should also be interpreted against its limitations. The final sample was a small group retained after study attrition which was likely biased toward cooperative parents. As is common in parenting studies, fathers were underrepresented, comprising less than 15% of the final sample. Previous research has similarly struggled to attain equality between parent genders, although fathers are potentially more likely to consider parenting programs if content is tailored to specific parenting situations [47]. Similarly, the sample had an overrepresentation of university-educated parents, again a common difficulty when recruiting for parenting studies, with 75% of the final sample reporting a Bachelor degree or higher. Nevertheless, with almost 90% of Australians owning a smartphone [86], there is potential to improve the reach of parenting programs and provide access to parents who would otherwise be unable to engage. Another possible limitation was that participants were required to finalise enrolment in the SEMA3 platform to receive EMA prompts. As another barrier towards participation, this may have been a step too far for some participants to remain engaged in the study. Additionally, due to the funding constraints of the study, only participants who completed the entirety of the baseline survey were considered for further participation, excluding participants who may have otherwise provided valuable EMA data. Finally, the current study did not offer parents any parenting resources or benefits aside from a supermarket voucher, potentially curtailing interest from groups which may have otherwise participated. It may be that a future parenting program designed as an ecological momentary intervention, with specific support tailored to families, may prove more attractive to parents to engage with.

## Conclusion

Parenting programs are an effective means of modifying parents' skills and behaviours and improving children's outcomes, but have struggled with low population reach and engagement. Online programs have offered potential to extend the reach and engagement of parenting programs, but so far have also struggled with low engagement and high attrition. The proliferation of smartphones amongst adults in Australia represents an opportunity for parenting interventions to provide timely and contextual resources to parents, but data is needed to guide intervention design. Findings from the current study provide concrete information about potential for tailoring parenting online and ecological momentary interventions to parents with young children. This study used a pre-existing set of short measures to develop a 1-minute ecological momentary assessment survey for parent and child negative affect and emotion dysregulation to then identify when parents were most likely to engage with smartphone prompts, when they were most often with their children, and the frequency at which different difficult parenting situations arose.

## Supporting information

**S1 Table. Short survey prompts schedules.**
(DOCX)

**S2 Table. Short survey response options for difficult parenting situations arranged by theme.**
(DOCX)

**S3 Table. Individual adult PANAS item regression results with unstandardised coefficients and 95% confidence intervals.**
(DOCX)

**S4 Table. Individual child PANAS item regression results with unstandardised coefficients and 95% confidence intervals.**
(DOCX)

**S5 Table. Individual adult S-DERS item regression results with unstandardised coefficients and 95% confidence intervals.**
(DOCX)

**S6 Table. Individual child S-DERS item regression results with unstandardised coefficients and 95% confidence intervals.**
(DOCX)

**S7 Table. Association of individual parent PANAS short survey items with baseline measures and subscales.**
(DOCX)

**S8 Table. Association of individual child PANAS short survey items with baseline measures and subscales.**
(DOCX)

**S9 Table. Association of individual parent S-DERS short survey items with baseline measures and subscales.**
(DOCX)

**S10 Table. Association of individual child S-DERS short survey items with baseline measures and subscales.**
(DOCX)

**S11 Table. Association of individual parent PANAS short survey items with other short survey items.**
(DOCX)

**S12 Table. Association of individual child PANAS short survey items with other short survey items.**
(DOCX)

**S13 Table. Association of individual parent S-DERS short survey items with other short survey items.**
(DOCX)

**S14 Table. Association of individual child S-DERS short survey items with other short survey items.**
(DOCX)

## Acknowledgments

The authors would like to acknowledge all the participants who offered their time and efforts to this project. In addition, Tomer S. Berkowitz would like to acknowledge Lauren and Liam for their help with code, Chris and Liam for providing specific code for some of the analysis, and Bonnie for all her work getting the project up and running.

## Author contributions

**Conceptualization:** Tomer S. Berkowitz, John W Toumbourou, Subhadra Evans, Elizabeth M Westrupp.

**Data curation:** Tomer S. Berkowitz, Elizabeth M Westrupp.

**Formal analysis:** Tomer S. Berkowitz, Elizabeth M Westrupp.

**Investigation:** Tomer S. Berkowitz, Elizabeth M Westrupp.

**Methodology:** Tomer S. Berkowitz, John W Toumbourou, Subhadra Evans, Matthew Fuller-Tyszkiewicz, Elizabeth M Westrupp.

**Project administration:** Tomer S. Berkowitz.

**Supervision:** John W Toumbourou, Subhadra Evans, Elizabeth M Westrupp.

**Writing – original draft:** Tomer S. Berkowitz, John W Toumbourou, Subhadra Evans, Elizabeth M Westrupp.

**Writing – review & editing:** Tomer S. Berkowitz, John W Toumbourou, Subhadra Evans, Matthew Fuller-Tyszkiewicz, Elizabeth M Westrupp.

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
