## [Decision Letter · Decision Letter 0]

PONE-D-24-39179Momentary assessment of parent and child emotion regulation to inform the design of a new emotion-focused parenting appPLOS ONE

Dear Dr. Berkowitz,

Thank you for submitting your manuscript to PLOS ONE. After careful consideration, we feel that it has merit but does not fully meet PLOS ONE’s publication criteria as it currently stands. Therefore, we invite you to submit a revised version of the manuscript that addresses the points raised during the review process.

We look forward to receiving your revised manuscript.

Kind regards,

Andrea Maugeri

Academic Editor

PLOS ONE

Journal Requirements:

Reviewers' comments:

Reviewer's Responses to Questions

**Comments to the Author**

1. Is the manuscript technically sound, and do the data support the conclusions?

Reviewer #1: Yes

Reviewer #2: Partly

2. Has the statistical analysis been performed appropriately and rigorously? 

Reviewer #1: Yes

Reviewer #2: Yes

3. Have the authors made all data underlying the findings in their manuscript fully available?

Reviewer #1: No

Reviewer #2: Yes

4. Is the manuscript presented in an intelligible fashion and written in standard English?

Reviewer #1: Yes

Reviewer #2: Yes

5. Review Comments to the Author

Reviewer #1: Thank you for providing the opportunity to review the manuscript. The study utilized the ecological momentary assessment method to gather data from Australian parents regarding outcomes related to parent and child emotion regulation and affect, which was an intriguing approach. The study results uncovered interesting patterns. However, the primary limitation of the study is the relatively small sample size, which hinders the generalizability of the findings. Please find below additional specific comments for your consideration:

1. Could you please provide information on the attrition rates reported for in-person parenting programs such as TIK and the Parent-Child Interaction Therapy-Emotion Development? Additionally, it would be valuable to include attrition rates for online parenting programs for comparison purposes.

2. (p.7, lines 145-147) “smoking cessation interventions aim to decrease or terminate smoking habits by providing a brief in-the-moment intervention that disrupts individuals’ urge to smoke.” Could you provide an example of a brief in-the-moment intervention strategy to illustrate this concept?

3. There is a typo on page 9, line 187. The comma at the end of the sentence should be a full-stop.

4. Given that the focus of this study is on utilizing EMA for collecting parenting data, it would be beneficial to delve into more detailed discussions of the two prior studies referenced (Dunton et al. and Li et al.). Can you elaborate on the protocols, sample sizes, and limitations of those studies? Why do the findings from those studies support the use of EMA for collecting in-the-moment responses from parents?

5. On page 10, lines 223-225, there is a mention of parents being allocated to schedules. Could you clarify at which stage of the flow chart shown in Figure 1 this allocation occurred? Additionally, please ensure that the tables are correctly ordered according to their reference numbers.

6. There appears to be a calculation error for the percentage in Figure 1 related to "Not enrolled in SEMA3 (n=25; 42%)."

7. On page 11, line 240, the phrase "and almost half the were aged 2 years" contains a redundant word ("the").

8. Regarding the five purpose-defined criteria mentioned, were these criteria developed by the research team or adopted from prior studies? It would be helpful to provide the rationale and justification for adopting these specific criteria.

9. Could you include the statistical results, such as regression outcomes, possibly as supplementary tables, to demonstrate how the scores in Table 4 for the five criteria were derived?

10. To enhance clarity in describing the results concerning the frequency of parent and child interactions, consider presenting the information in terms of specific time periods rather than general descriptors like "during the morning," "early evening," etc.

11. Table 10 and the preceding discussion regarding parents' preferences for tailored parenting interventions seems somewhat disconnected from the overall flow of the discussion. It may benefit from further integration with the study's findings to avoid repetition of the discussion in the introduction.

12. While the study demonstrates the feasibility of using EMA to collect parenting data, further elaboration is needed on how this relates to EMA as an engaging approach for parenting interventions, with more in-depth justifications.

Reviewer #2: The present study sought to inform the development of an emotion-focused parenting program designed to provide in-the-moment parenting support. While I fully agree that parenting programs would benefit from smartphone apps designed to extend reach and engagement, the methods of the present study had some major weaknesses that limit the findings.

1) One concern is the recruitment method. The authors recruited via social media ads. The use of social media might bias the results by focusing on parents who are more engaged in mobile apps relative to the general population. If the goal of the EMI app is to extend the reach of parenting programs, then the methods should be tested with a wide range of parents, including those not as engaged in social media and smartphone use. Also, the sample seems skewed toward the highly educated. Many parenting programs are geared toward low-income or lower-educated samples who may be vulnerable to experiencing parenting stressors. How does the present sample represent that population?

2) How were the three parents in the final sample who were excluded for not providing any EMA data different from the 47 parents excluded due to not completing any EMA surveys?

3) The authors state that participants were primarily excluded due to issues of ineligibility (e.g., not residing in Australia). However, based on Figure 1, the most common reason for exclusion was not completing the baseline survey. Why was a complete baseline survey required for study participation? It seems like excluding all those who failed to complete the baseline survey would bias the sample towards the most compliant participants, but not give a comprehensive account of parental behavior with the app (e.g., those who are less compliant were excluded, which was 40% of all potential participants). Similarly, another 25 parents were excluded for not enrolling in SEMA3. This again seems to be biasing the results by focusing on the most compliant and engaged parents. Therefore, results of the 89 participants may suggest the “optimal (1) item selection for measuring in-the-moment parent and child negative affect and emotion dysregulation, (2) timing of prompts, (3) selection of daily experiences of difficult parenting situations in an Australian setting, as well as (4) establish times of day and parenting situations that precipitate heightened levels of parent and child negative affect and emotion dysregulation.” However, these results will be based on the “ideal” EMA participant. Perhaps those that were excluded had more difficulty with emotion regulation and, therefore, results would have differed had more of them been included.

4) Somewhat relatedly, the percentages in Figure 1 are confusing. How is 25 participants 42% when the denominator is seemingly 184?

5) Have all the child measures been validated for children as young as 2-4? As an example, I believe the Short Mood and Feelings Questionnaire is for children 6+.

6) Why only include negative affect and emotion dysregulation? When trying to understand parent/child emotion and regulation, it seems important to also capture positive emotions and adaptive regulation. Relatedly, the selected items (bolded items from Table 4) reflect the most basic and common emotions/regulation challenges for children this age. Specifically, given that these children are 2-4 it is highly normative for the parents to indicate that the children have difficulty controlling their behavior and are often mad or sad. It’s not clear what the study actually taught us besides confirming existing knowledge that mad and sad are basic and common emotions and that the question “I (or my child) have difficulty controlling my (their) emotions” is a reliable measure of emotion dysregulation.

7) It is unclear how the results of the present study will be used to inform the EMI parent app. Are the authors arguing that, due to the results of the present study, they now know to conduct brief interventions towards the beginning of the day and at the end of the day? The field of EMA has already established these best practices, so it’s unclear what was learned from the study.

8) Some minor editing for typos and grammatical errors is needed.

6. PLOS authors have the option to publish the peer review history of their article (what does this mean? ). If published, this will include your full peer review and any attached files.

**Do you want your identity to be public for this peer review?** For information about this choice, including consent withdrawal, please see our Privacy Policy .

Reviewer #1: No

Reviewer #2: No

---

## [Author Response · Author response to Decision Letter 1]

11 May 2025

REVIEWER 1

Section 5, Comment 1: Could you please provide information on the attrition rates reported for in-person parenting programs such as TIK and the Parent-Child Interaction Therapy-Emotion Development? Additionally, it would be valuable to include attrition rates for online parenting programs for comparison purposes.

Attrition rates for in-person parenting programs have been added to the manuscript (see lines 101-103):

"An early community trial of Tuning in to Kids reported a 17-20% attrition rate for the intervention sample, while an RCT of Parent-Child Interaction Therapy-Emotion Development reported a 30% attrition rate."

Attrition rates for online parenting programs have been added to the manuscript (see lines 113-114):

"…with short-term follow up attrition rates 24% and long-term follow up rates 36%."

Section 5, Comment 2: (p.7, lines 145-147) “smoking cessation interventions aim to decrease or terminate smoking habits by providing a brief in-the-moment intervention that disrupts individuals’ urge to smoke.” Could you provide an example of a brief in-the-moment intervention strategy to illustrate this concept?

Examples of in-the-moment intervention strategies have been added to the manuscript (see lines 150-152):

"Interventions can include strategies such as watching videos, meditation, tips, and supportive messages."

Section 5, Comment 3: There is a typo on page 9, line 187. The comma at the end of the sentence should be a full-stop.

Thank you for identifying this mistake – the typographical error has been corrected.

Section 5, Comment 4: Given that the focus of this study is on utilizing EMA for collecting parenting data, it would be beneficial to delve into more detailed discussions of the two prior studies referenced (Dunton et al. and Li et al.). Can you elaborate on the protocols, sample sizes, and limitations of those studies? Why do the findings from those studies support the use of EMA for collecting in-the-moment responses from parents?

Additional information has been added regarding Dunton et al. and Li et al, as well as a brief justification for the use of EMA in parenting contexts (see lines 186-192):

"Dunton et al.’s study recruited 199 mothers of children aged 8-12 to complete two 7-day periods of EMA data collection, with survey prompts sent up to eight times per day. Li and Lansford recruited 184 parent-child dyads of children with and without ADHD and prompted participants to answer one questionnaire a day for 7 days. Although both studies stated limitations due to participant self-report, both utilised EMA via smartphones, strengthening support for its use in parenting contexts."

Section 5, Comment 5: On page 10, lines 223-225, there is a mention of parents being allocated to schedules. Could you clarify at which stage of the flow chart shown in Figure 1 this allocation occurred? Additionally, please ensure that the tables are correctly ordered according to their reference numbers.

Figure 1 has been updated with schedule allocation and the tables have been relabelled in their correct order.

Section 5, Comment 6: There appears to be a calculation error for the percentage in Figure 1 related to "Not enrolled in SEMA3 (n=25; 42%)."

Thank you identifying this mistake – Figure 1 has been updated with the correct percentage for “Not enrolled in SEMA3”.

Section 5, Comment 7: On page 11, line 240, the phrase "and almost half the were aged 2 years" contains a redundant word ("the").

Thank you identifying this mistake – the typographical error has been corrected.

Section 5, Comment 8: Regarding the five purpose-defined criteria mentioned, were these criteria developed by the research team or adopted from prior studies? It would be helpful to provide the rationale and justification for adopting these specific criteria.

The criteria were developed by the research team. As the current study is informing the development of a larger parenting program, specific criteria were considered to align with the intended aims of the parenting program. This has now been elaborated upon (see lines 281-283):

"As the current study informs a larger project seeking to develop a parenting program as a smartphone app, these criteria were decided as being relevant to the overall goals."

As explained in the manuscript (lines 441-443), there are no guidelines explicitly governing the development of EMA surveys, although different approaches have been used previously (Forsell et al., 2019; Jimenez et al., 2022; Rosenkranz et al. 2020). In particular, Forsell et al. (2019) and Jimenez et al. (2022) emphasised the need to align EMA survey items with study aims and EMA methodology.

Section 5, Comment 9: Could you include the statistical results, such as regression outcomes, possibly as supplementary tables, to demonstrate how the scores in Table 4 for the five criteria were derived?

The statistical results for Sensitivity, Relevance, Alignment, and Frequency have been included as supplementary tables (noting that Validity was ranked by the research team’s assessment of face validity, and as such does not have a statistical output; see S3 Table-S14 Table).

Section 5, Comment 10: To enhance clarity in describing the results concerning the frequency of parent and child interactions, consider presenting the information in terms of specific time periods rather than general descriptors like "during the morning," "early evening," etc.

The description of results for Table 6 has been updated to improve clarity by also specifying time periods (see line 394-395):

"Children were most often with their parents during the morning, specifically 7:30am, and early evening (6:00pm)."

Section 5, Comment 11: Table 10 and the preceding discussion regarding parents' preferences for tailored parenting interventions seems somewhat disconnected from the overall flow of the discussion. It may benefit from further integration with the study's findings to avoid repetition of the discussion in the introduction.

Table 10 has been deleted and its key points integrated into a new paragraph (see response to next comment for further elaboration).

Section 5, Comment 12: While the study demonstrates the feasibility of using EMA to collect parenting data, further elaboration is needed on how this relates to EMA as an engaging approach for parenting interventions, with more in-depth justifications.

We acknowledge and agree that further elaboration is needed to explain why EMA is an encouraging method for use with parenting programs. As such, a new paragraph has been added to further justify why EMA is an engaging and useful approach for parenting programs (see lines 533-541):

"In considering whether to use EMA for parenting programs, the current study examined whether a smartphone-based approach would result in higher levels of engagement by parents and how programs could maximise the efficacy of their content. Our findings offer concrete information on how similar designs could be implemented within a mental health prevention context. In understanding the barriers that families face with in-person programs, online approaches can help to mitigate some of these obstacles whilst reaching families that are most substantially limited by them. In turn, with parents indicating their desire for content that more accurately reflects their own personal circumstances, providing opportunities for support and learning both when parents can access them and in times when their needed and convenient may precipitate stronger engagement and improved outcomes. Parents expressed a desire for content that accurately represents their circumstances, with flexible access to support and learning opportunities. Our findings align with evidence from Finan et al [48], Hansen, Broomfield and Yap [46], and Sicouri et al [47], indicating that parents prefer programs tailored to their context. The diversity in reported parenting challenges – both in type and timing – suggests that future programs should reflect a wide range of family situations. Ensuring availability when needed and at convenient times may enhance engagement and improve outcomes."

REVIEWER 2

Section 5, Comment 1: One concern is the recruitment method. The authors recruited via social media ads. The use of social media might bias the results by focusing on parents who are more engaged in mobile apps relative to the general population. If the goal of the EMI app is to extend the reach of parenting programs, then the methods should be tested with a wide range of parents, including those not as engaged in social media and smartphone use. Also, the sample seems skewed toward the highly educated. Many parenting programs are geared toward low-income or lower-educated samples who may be vulnerable to experiencing parenting stressors. How does the present sample represent that population?

We acknowledge that recruitment via social media likely resulted in a non-representative sample. While research shows that many parents use platforms like Facebook for parenting information (Frey et al., 2022; Bennetts et al., 2019; Archer & Kao, 2018; Hendry et al., 2024), we recognise that social media cannot reach all parents.

Section 5, Comment 2: How were the three parents in the final sample who were excluded for not providing any EMA data different from the 47 parents excluded due to not completing any EMA surveys?

Apologies, we realise that this was not clearly described in the manuscript. We have now updated our description to clarify that these two groups of parents were excluded at different stages (see lines 241-245):

"…with a final 47 excluded for not logging into the SEMA3 app and therefore not receiving any EMA surveys. After exclusions, the final sample consisted of 89 parents with full baseline data who engaged with the SEMA3 platform. Three parents included in the final sample received EMA prompts but did not complete any EMA surveys."

Section 5, Comment 3: The authors state that participants were primarily excluded due to issues of ineligibility (e.g., not residing in Australia). However, based on Figure 1, the most common reason for exclusion was not completing the baseline survey. Why was a complete baseline survey required for study participation? It seems like excluding all those who failed to complete the baseline survey would bias the sample towards the most compliant participants, but not give a comprehensive account of parental behavior with the app (e.g., those who are less compliant were excluded, which was 40% of all potential participants). Similarly, another 25 parents were excluded for not enrolling in SEMA3. This again seems to be biasing the results by focusing on the most compliant and engaged parents. Therefore, results of the 89 participants may suggest the “optimal (1) item selection for measuring in-the-moment parent and child negative affect and emotion dysregulation, (2) timing of prompts, (3) selection of daily experiences of difficult parenting situations in an Australian setting, as well as (4) establish times of day and parenting situations that precipitate heightened levels of parent and child negative affect and emotion dysregulation.” However, these results will be based on the “ideal” EMA participant. Perhaps those that were excluded had more difficulty with emotion regulation and, therefore, results would have differed had more of them been included.

Thank you for bringing this ambiguity to our attention. Both these points regarding participant exclusion have been additionally addressed in the limitations section.

Of participants who were excluded due to incomplete baseline responses, 95% completed no more than the demographic items, i.e., they did not provide responses to any of the measures. Prior to data collection, the research team decided that participants were required to complete the baseline survey in its entirety so that data could be used for analysis in another study. As there were limited resources to reimburse participants for their time, only participants with fully complete baseline surveys were considered for enrolment in the EMA stage (see lines 568-571):

"Additionally, due to the funding constraints of the study, only participants who completed the entirety of the baseline survey were considered for further participation, excluding participants who may have otherwise provided valuable EMA data."

Participants who were excluded for not enrolling in SEMA3 were invited to enrol in the program, however, as they did not proceed with enrolment on their end, they were unable to receive survey prompts and were therefore not able to complete surveys (see lines 565-568):

"Another possible limitation was that participants were required to finalise enrolment in the SEMA3 platform to receive EMA prompts. As another barrier towards participation, this may have been a step too far for some participants to remain engaged in the study."

Section 5, Comment 4: Somewhat relatedly, the percentages in Figure 1 are confusing. How is 25 participants 42% when the denominator is seemingly 184?

Thank you identifying this mistake – the numerical error has been corrected (it now reads 14%).

Section 5, Comment 5: Have all the child measures been validated for children as young as 2-4? As an example, I believe the Short Mood and Feelings Questionnaire is for children 6+.

Not all the measures have been validated for children aged 2-4; however, we were not able to find equivalent measures that could be used with younger children. As such, measures whose items appeared appropriate for children aged 2-4 years were used (see lines 260-262):

"Although not all measures were validated for use with children aged 2-4 years, measures whose items were deemed appropriate for the age range were selected."

Section 5, Comment 6:Why only include negative affect and emotion dysregulation? When trying to understand parent/child emotion and regulation, it seems important to also capture positive emotions and adaptive regulation. Relatedly, the selected items (bolded items from Table 4) reflect the most basic and common emotions/regulation challenges for children this age. Specifically, given that these children are 2-4 it is highly normative for the parents to indicate that the children have difficulty controlling their behavior and are often mad or sad. It’s not clear what the study actually taught us besides confirming existing knowledge that mad and sad are basic and common emotions and that the question “I (or my child) have difficulty controlling my (their) emotions” is a reliable measure of emotion dysregulation.

As the study is seeking to inform a parenting program that improves parent and child emotion regulation, items that measured negative affect and emotion dysregulation were important to capture. The program will aim to improve the capacity for parents to improve their own emotion regulation when dealing with their chil

---

## [Editor Report · Decision Letter 1]

Momentary assessment of parent and child emotion regulation to inform the design of a new emotion-focused parenting app

PONE-D-24-39179R1

Dear Dr. Berkowitz,

We’re pleased to inform you that your manuscript has been judged scientifically suitable for publication and will be formally accepted for publication once it meets all outstanding technical requirements.

Kind regards,

Andrea Maugeri

Academic Editor

PLOS ONE
---

## [Editor Report · Acceptance letter]

PONE-D-24-39179R1

PLOS ONE

Dear Dr. Berkowitz,

I'm pleased to inform you that your manuscript has been deemed suitable for publication in PLOS ONE. Congratulations! Your manuscript is now being handed over to our production team.

Kind regards,

on behalf of

Dr. Andrea Maugeri

Academic Editor

PLOS ONE